



**Impact of elevated precipitation, nitrogen deposition and warming on soil**
**respiration in a temperate desert**
Ping Yue[1,2,3], Xiaoqing Cui[2], Yanming Gong[1], Kaihui Li[1], Keith Goulding[4], Xuejun Liu[2] [*]
[1] State Key Laboratory of Desert and Oasis Ecology, Xinjiang Institute of Ecology and Geography,
Chinese Academy of Sciences, Urumqi 830011, China
[2] Key Laboratory of Plant-Soil Interactions of MOE, College of Resources and Environmental
Sciences, China Agricultural University, Beijing 100193, China
[3] University of the Chinese Academy of Sciences, Beijing 100039, China
[4] The Sustainable Soils and Grassland Systems Department, Rothamsted Research, Harpenden
AL5 2JQ, UK
* Correspondence to: Xuejun Liu (liu310@cau.edu.cn; or ecology2100@sina.cn)
**Abstract**
Soil respiration ($R_s$) is the most important source of carbon dioxide emissions from soil to
atmosphere. However, it is unclear what the interactive response of $R_s$ would be to environmental
changes such as elevated precipitation, nitrogen (N) deposition and warming, especially in unique
temperate desert ecosystems. To investigate this an *in situ* field experiment was conducted in the
Gurbantunggut Desert, northwest China, from September 2014 to October 2016. The results
showed that precipitation and N deposition significantly increased $R_s$, but warming decreased $R_s$,
which was mainly through its impact on the variation of soil moisture at 5 cm depth. In addition,
the interactive response of $R_s$ to combinations of the factors was much less than that of any
single-factor, and the main interaction being a positive effect, except interaction from increased



precipitation and high N deposition (60 kg N ha$^{-1}$ yr$^{-1}$). Although R$_s$ was found to be a unimodal
change pattern with the variation of soil mositure, soil temperature and soil NH$_4^+$-N content, and it
was signicantly postively correlated to soil dissloved organic carbon (DOC) and pH, but from a
structural equation model found that soil temperature was the most important controlling factor.
Those results indicated that R$_s$ was mainly interactively controlled by the soil multi-environmental
factors and soil nutrients, and was very sensitive to elevated precipitation, N deposition and
warming. But the interactions of multiple factors largely reduced between-year variation of R$_s$
more than any single-factor, suggesting that the carbon cycle in temperate deserts could be
profoundly influenced by positive carbon-climate feedbacks.
**Key words**: precipitation; nirogen deposition; warming; soil respiration; temperate desert

**Highlights**
1. Impacts of rainfall, N addition and warming on R$_s$ were studied in a temperate desert.
2. Rainfall and N deposition significantly increased R$_s$, but warming reduced it.
3. The interactive response of R$_s$ was much lower than any single-factor.
4. Soil temperature was the most important controlling factor for R$_s$.

**1. Introduction**

Global climate warming, changes in precipitation patterns and increased atmospheric

nitrogen (N) deposition have all occurred since the industrial revolution, especially in temperate
regions (IPCC, 2013), which will be expected to significantly change soil respiration (R$_s$) that is
the most important source of carbon dioxide (CO$_2$) from soil to atmosphere (Wu et al., 2011): the



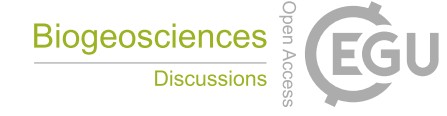

annual $CO_2$ flux from $R_s$ was ten-fold that of fossil fuel emissions (Eswaranet al., 1993; Batjes,
1996; Gougoulias et al., 2014). Therefore, even a small change in $R_s$ will profoundly affect
greenhouse gas balance and climate (Heimann and Reichstein, 2008). Although a number of
experiments of the effects of warming, precipitation, and N deposition on $R_s$ have been conducted
in alpine grassland, tundra regions, peatlands and temperate forest (Lafleur and Humphreys, 2008;
Strong et al., 2017; Yang et al., 2017; Zhaoet al., 2017), studies in temperate desert ecosystems are
scarce, especially the impact on $R_s$ of the interactions of these changes. A field study of
multi-factor interactive effects on $R_s$ was therefore conducted in a temperate desert ecosystem to
help in understanding the response of $R_s$ to climate change and N deposition in future and
highlight the main driving factors.

$R_s$ includes autotrophic respiration ($R_A$), which is mainly from plant roots and mycorrhizal

activities; and heterotrophic respiration ($R_H$), which is mainly from the activities of
microorganisms (Hanson et al., 2000). Soil moisture is an critical limiting factor for plant roots
and microbial activities in desert ecosystems (Huang et al., 2015a): $R_s$ was significantly increased
by 47-70% in a degraded steppe in Inner Mongolia, China, by increasing precipitation (Chen *et al.*,
2013), with the effect especially strong in summer (Zhang et al., 2017). In addition, in arid
ecosystems, increasing precipitation significantly stimulated plant growth, enhanced soil microbial
activity and abundance (Huanget al., 2015a), and changed soil nutrient and substrate concentration,
such as dissloved organic carbon (DOC), inorganic nitrogen content, moisture and temperature
(Huang et al., 2015b).

Warming significantly increased soil temperature, another important controlling factor for

plants growth and microbial activity (Sheiket al., 2011; Huang et al., 2015a). $R_s$ rates were



significantly increased in a forest soil and Tibetan Plateau grassland by warming (Chen et al.,
2017a), reducing $R_s$ with decreasing soil moisture in the growing season, but increasing $R_s$ in the
non-growing season (Fang et al., 2017; Li et al., 2017); no significant impact was observed from
warming (Liu et al., 2016a). Therefore, how $R_s$ is affected by warming induced variations in the
soil environment is still unclear. In addition, low and short-term N deposition enhanced $R_s$, while
higher and long-term N deposition inhibited $R_s$ due to changes in plant growth and microbial
activity (Zhu et al., 2017), but no impacts have also been reported (Luo et al., 2017; Zhang et al.,
2017). A meta-analysis showed that the effects of N enrichment on soil $CO_2$ fluxes depended on
temperature and soil properties (Zhong et al., 2016); desert soils may be even more senstive to its
variation.

A nation-wide analysis showed that warming, elevated N deposition and precipitation

significantly increased $R_s$ in China (Feng et al., 2017). Some studies have shown that the warming
effect on $R_s$ mainly depended on the variation of soil moisture in a dry forest soil (Li et al., 2017).
Luo et al. (2008), using a modeling analysis, found that interactive effects became increasingly
weaker with increasing intensity of the factors, but a recent meta-analysis showed that interactive
effects were much greater than single factors (Zhou et al., 2016a). Thus how multi-factor
interactions impact $R_s$ is still unclear. Therefore, an *in stiu* experiment was carried out in the
Gurbantunggut Desert to (1) investigate the single-factor and interactive responses of $R_s$ to
warming, precipitation and N deposition, and (2) identify the main controlling factors on $R_s$.

**2.  Materials and methods**
*2.1. Study site*





A field experiment was carried out at the southern edge of the Gurbantunggut Desert (44º26'
N, 87 54'E and 436.8 m a.s.l.), northwest China, from September 2014 to October 2016. This is
the largest fixed/semi-fixed temperate desert in China. The mean annual temperature and
precipitation are 7.1 ℃ and 215.6 mm, respectively (Cui et al., 2017), and annual potential
evaporation exceeds 2000 mm. From late November to mid-March of the following year, a 20–35
cm depth of snow cover the whole desert (equivalent to 38–64 mm rainfall;Huang et al., 2015c).
The growing season is from April to October. This desert soil is of extremely low fertility and high
alkaline (Cui et al., 2017). Soil organic carbon, total N content, soil $NO_3^-$-N, $NH_4^+$-N contents and
C:N ratio are 2.21 ±0.71 g $kg^{-1}$, 0.08 ±0.003 g $kg^{-1}$, 4.49 ±0.71 mg $kg^{-1}$, 1.38 ±0.74 mg $kg^{-1}$ and
21.39 ± 1.84, respectively (Table 1; Cui et al., 2017). Plant species are dominated by *Haloxylon*
*ammodendron* and *Haloxylon persicum*, and the vegetation was extremely sparse, with only 30%
coverage, with some spring ephemeral plants (May–June), some annuals, and perennials
herbaceous plants (July–August; Liu et al., 2016). Spring ephemerals account for > 60% of the
community cover and 85% of the biomass. Summer ephemerals, annuals and perennials usually
account for only a small proportion of the community biomass before June, but dominate the
community after the die-back of the spring annuals (Huang et al., 2015c).

*2.2. Experimental treatments*
A striking N deposition rate (35.2 kg N $ha^{-1}$ $yr^{-1}$) has occurred in the Gurbantunggut Desert
due to the rapid development of agriculture and industry with main form of ammonium nitrate
($NH_4NO_3$), and wet (19.6 kg N $ha^{-1}$ $yr^{-1}$) and dry (15.6 kg N $ha^{-1}$ $yr^{-1}$) deposition are almost half
(Song et al., 2015). In addition, according to the forecast of Galloway et al. (2008) that



atmospheric N deposition will double from the early 1990s to 2050, and the predictions of Liu et
al. (2010) that precipitation in this region would be increased by 30% in next 30 years. In June
2014, an *in situ* complete block interactive experiment was therefore conducted to study the
impact of N deposition and increased precipitation on $R_s$ (Experiment 1). The three levels of N
deposition (0 kg N ha$^{-1}$ yr$^{-1}$ (control, N0), 30 kg N ha$^{-1}$ yr$^{-1}$ (low, N1) and 60 kg N ha$^{-1}$ yr$^{-1}$ (high,
N2)) and two levels of precipitation ('natural' precipitation (W0) and an increase of 30% (an extra
60 mm precipitation annually (W1)) were applied (Cui et al., 2017). Therefore there were six
treatments (W0N0, W0N1, W0N2, W1N0, W1N1 and W1N2) with four replicates of each
treatment; each replicate plot was 10 m × 10 m with a 5-m wide buffer zone. The additional
precipitation and N deposition (NH$_4$NO$_3$) were added twelve times in April, July and September,
equivalent to 5 mm precipitation and 2.5 or 5 kg N ha$^{-1}$ per application over a week. The NH$_4$NO$_3$
was diluted in 50 L water (equal to 0.5 mm precipitation), and evenly applied following the
simulated precipitation. The same amount of water was applied to the control plots (W0N0).
Rapidly warming (0.6 ºC per decade), increasing precipitation (3-5 mm yr$^{-1}$ since 1979) and
receiving high N deposition (3 kg N ha$^{-1}$ since 1980) are affecting the Gurbantunggut Desert (Liu
et al., 2013; Li et al., 2015), which would be excepted to affect rate of $R_s$. Therefore, another
interactive experiment was established at the same time, simulating the three most likely climate
scenarios in the future: (1) warming only (W0N0T1); (2) increased precipitation and N deposition
without warming (W1N1T0); (3) the interaction of increasing precipitation, N deposition and
warming (W1N1T1); all compared with the current climate (W0N0T0). Therefore, there were four
treatments (W0N0T1, W1N1T0, W1N1T1, W0N0T0) with four replicates (plots) of each
treatment. Open-top chambers (OTCs) were used to simulate warming. The OTCs were designed



with 5 mm transparent tempered glass and stainless steel angle iron to the ITEX standard (Marion
et al., 1997). They were 2 m high and 4 m in diameter, with each OTC area being 12 m$^2$. However,
the design was improved such that the top and bottom OTC areas were the same so that
precipitation and snowfall were the same as that to the surrounding environment; this also avoids
overheating inside the OTCs. The timings of applications of water and N were as in Experiment 1.

*2.3. Measurements*

R$_s$ in all plots were measured twice or thrice a week (continuous measurements over 3 days

were made following simulated precipitation and N deposition) using gas chromatography and
static chambers (50 cm×50 cm×10 cm) at locations where grow only spring ephemeral plants
without any annuals and perennials in order to minimize the between-treatment spatial
heterogeneity due to sparse annuals, and perennials (Liu et al., 2012). Gas samples were collected
between 10:00-12:00 (GMT + 8) throughout the experimental period, which was detected in this
period were close to the diurnal averages (Fig.3b and 3d). Gas samples were collected from the
headspace of each chamber 0, 10, 20 and 30 min after closing the chamber per time. The gas
samples analyzed within three days using a gas chromatograph (GC; Agilent 7890A, Agilent
Technologies, Santa Clara, CA, USA) equipped with a flame ionization detector for quantitative
R$_s$ (Liu et al., 2012). R$_s$ rates were calculated from four concentrations of the gas sample based on
a first order differential linear or non-linear equation and were temperature- and pressure-
corrected (Liu et al., 2012; Zhang et al., 2014). Soil samples were taken monthly from around the
static chambers to a depth of 10 cm using an auger (3.5 cm in diameter). Fine roots and small
stones were separated out using a 2 mm sieve. Dissolved organic carbon (DOC) was extracted




with deionized water (soil: water ratio = 1:10) by shaking on an orbital shaker at 10000 rpm for 5
min and analyzed using a TOC analyzer (multi N/C 3100, Jena, Germany; Jones and Willett,
2006). Brookes' (1985), Chloroform fumigation extraction was used to measure microbial
biomass carbon (MBC) and microbial biomass nitrogen (MBN). Soil organic carbon (SOC) and
pH were measured using the potassium dichromate method (Jiang et al., 2014), and soil $NO_3^-$-N
and $NH_4^+$-N analyzed as per Yue et al. (2016). Caipos Soil and Environment Monitoring Systems
(Caipos GmbH, Austria) were used to monitor soil moisture/temperature at 5 and 20 cm depth
every hour.

*2.4. Effects of each treatment on $R_s$*
The each treatment effect was analyzed using the following formula to better evaluating the
effect of precipitation, warming and N deposition on $R_s$ (Yue et al., 2016).
$$\text{The treatment effect} = (TR_s - CR_s) / CR_s \times 100\%,$$
Where the treatment effect is W0N1, W0N2, W1N0, W1N1, W1N2, W1N1T1 or W0N0T1
effect on $R_s$ (a positive value shows that the treatment has increased $R_s$ and a negative value shows
decrease of $R_s$), corresponding $TR_s$ represents $R_s$ from the W0N1, W0N2, W1N0, W1N1, W1N2,
W1N1T1 or W0N0T1 plots (mg C $m^{-2}$ $h^{-1}$) and $CR_s$ indicates the $R_s$ from the control plots (W0N0,
mg C $m^{-2}$ $h^{-1}$).

*2.5. Statistical analyses*
Treatments effect on soil organic carbon (SOC), $NO_3^-$-N, $NH_4^+$-N content, pH, DOC, MBC
and MBN were examined in each treatment by least significant difference LSD ($p<0.05$). The



single-factor and interaction effects of precipitation, warming and N deposition on $R_s$ were
detected by multi-way analysis of variance (ANOVA), and the accumulated effect of precipitation,
warming and N deposition on $R_s$ were tested by repeated measures ANOVA. In addition, the
relationships of $R_s$ and DOC, MBC, MBN, soil temperature, soil moisture, $NH_4^+$-N content, soil
$NO_3^-$- N, and pH were described using a linear or non-linear regression model. The factors of key
controls on $R_s$ were used to analyze by structural equation models (SEMs). SPSS software
(version 20.0) was used to conduct all statistical analyses, and statistical significant differences
were set with $P<0.05$. All Figures were created using the Sigmaplot software package (version
10.0), but SEMs analyses were carried out using AMOS 22.0 (Amos Development Corporation,
Chicago, IL, USA).

**3.  Results**
*3.1. Treatments effects on soil environmental and properties*

Soil temperatures at 5 depth were mostly increased between 11:00 and 22:00 every day by

warming; the average annual soil temperatures at 5 and 20 cm depth were significantly increased
by 4.41 and 3.67 $^0$C, respectively (Fig. 1a). Soil moisture at 5 cm depth was decreased by warming
by only 0.61v/v% (Fig. 1b), and a very small decrease of 0.01v/v% in soil moisture at 20 cm depth
was observed (Fig.1b). Soil moisture at 5 and 20 cm depth were largely increased by the
interaction of precipitation, N deposition and warming (Fig.1b). N deposition and warming
significant increased soil $NH_4^+$-N and $NO_3^-$-N contents (Fig. 1c), but no significant change was
found from increased precipitation. Soil MBC and MBN were greatly increased by N deposition,
but significant negative effects on soil MBC and MBN were observed by warming and the



interaction of precipitation and N deposition (Fig. 1d). No significant change in SOC and DOC
was observed in any treatment (Fig. 1c and 1d).

*3.2. Precipitation, warming and N deposition effects on* $R_s$
In our study, a weak $R_s$ emission rate (-2.46 to 50. 26 mg C m$^{-2}$ h$^{-1}$) was observed at control
plots with an average emission rate of 12.18 mg C m$^{-2}$ h$^{-1}$ from September 2014 to October 2016
(Fig. 2b). Annual cumulative rate of $R_s$ was 1090.11±450.78 kg C ha$^{-1}$, with non-growing season
account for 20.7% of the annual emission (Table 1). $R_s$ was significantly enhanced by increasing
5-mm precipitation and N deposition from 12.18 to 16.23 and 14.97 mg C m$^{-2}$ h$^{-1}$ (average),
respectively (P< 0.001; Fig. 2b and 2c; Table 2), with annual $R_s$ increased by 33.1% and
19.2-22.8%, respectively (Table 1). And the low N deposition effect on $R_s$ was much higher than
that high N deposition (Fig. 2b and 2c). However, $R_s$ was reduced mostly by warming, although
not significant (P=0.084; Table 2). And high temperatures at times of peak sunshine during the
diurnal variation significantly inhibited its emission rate (Fig 3a and 3b), but it was also
significantly increased by warming following rainfall that increased soil moisture (Fig. 3c and 3d).
The diurnal trend in $R_s$ was consistent with that of soil temperature at 5 cm depth (Fig. 3). In
addition, the interactive responses of $R_s$ to increasing precipitation, warming and N deposition
were much lower than that from any single-factor (Table 1), and with the interaction of 60 kg ha$^{-1}$
N and extra precipitation decreasing $R_s$ by 4.25% (Table 1). Overall, annual $R_s$ rates were
significantly impacted by precipitation, N deposition, and their interaction (Table 2), but no
significant net change was caused by warming (Table 2), although $R_s$ rates were decreased by 9.99%
(Table 1).




*3.3. Temporal variation and its control*

224 The results of repeated measures ANOVA showed that significantly accumulated effects on

225 $R_s$ were found by N deposition and interaction between N deposition and precipitation or warming

226 rather than alone increasing precipitation and warming (Table 2). A large between-year variation

227 in $R_s$ was observed with a coefficient of variation (CV) up to 41.4% (a much higher $R_s$ rate was

228 observed in 2016 than 2015), but variation was reduced by increasing precipitation, N deposition

229 and warming and their interaction, except with an increase in N deposition of 30 kg ha$^{-1}$ (Table 1).

230 The results of regression analysis showed that $R_s$ was significantly increased by increases in pH

231 and DOC (Fig. 4e and 4f), but no significant relationships were found with MBC, MBN or

232 $NO_3^-$-N content (Fig. 4a, 4b and 4c). In addition, thresholds in the impacts of increased soil

233 mositure, soil temperature and $NH_4^+$-N content on $R_s$ were found. Soil mositure was the most

234 important controlling factor when it was <2% or soil temperature was >37$^0$C (Fig. 4g and 4h).

235 Secondly, soil temperature was the most important limiting factor when soil moisture was >20%

236 or soil temperature <12 $^0$C (Fig. 4g and 4h). Thirdly, there was no significant impact on $R_s$ when

237 soil $NH_4^+$-N content was <6.3 mg N / kg. A significant increase in $R_s$ occurred when soil $NH_4^+$-N

238 content was between 6.3 and 12.6 mg N / kg, but $R_s$ was inhibited when soil $NH_4^+$-N content was

239 between 12.6 and 31.6 mg N / kg (Fig. 4d).


241 **4. Discussion**

242 *4.1. Single-factors impacts of precipitation, N deposition and warming on $R_s$*

243 Annual $R_s$ was 1090 kg C ha$^{-1}$ in this temperate desert, with the non-growing season

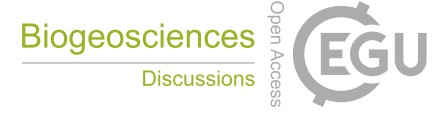



accounting for 20.7 % of the annual flux (Table 1). This is consistent with previous study in here
(Zhou et al., 2014; Huang et al., 2015a) because SOC content was very low (Fig. 1c), and
vegetation was very sparse in this desert (Liu et al., 2016b). Increasing precipitation significantly
increased $R_s$ (Fig. 2b). It is also consistent with the results of a meta-analysis and previous study in
here (Huang et al., 2015a ; Liu et al., 2016c). This is because that the growth of desert plants and
microbial activity are significantly activated by increasing precipitation (Huang et al., 2015a), and
microbial biomass, mass-specific respiration, microbial biomass carbon (MBC) and nitrogen
(MBN), and microbial PLFAs were consistently significantly enhanced by increased precipitation
(Zhang et al., 2013; Huang et al., 2015a). However, $R_s$ in our study was much higher in moderate
soil moisture conditions than with too little or too much soil moisture (Fig. 4g). This suggests that
$R_s$ is mainly $R_H$ rather than $R_A$ in this desert, namely from soil microorganism, because (1) too
little or too much soil moisture could significantly inhibit microbial activity due to variation of
soil temperature and soil properties (Ma et al., 2013), while moderate soil moisture could
significantly enhance microbial activity (Skopp et al., 1990), and (2) the biomass of fine roots was
no significantly enhanced at our sites by increased precipitation (Cui et al., 2017). This is
consistent with results from a desert steppe in northern China where the contribution of $R_H$ (78.1%)
was significantly higher than that of $R_A$ (21.9%) under increasing precipitation (Liu et al., 2016a).

N deposition also significantly increased $R_s$, especially in low N deposition (Fig. 2c). This is

consistent with results from an alpine meadow and in the Loess Plateau (Fang et al., 2017; Zong et
al., 2017), and with a meta-analysis showing that N deposition increased $R_s$ by 8.8% (Zhou et al.,
2016a). This is because N deposition, on the one hand, could increase fine root biomass, although
this was not significant in our study (Cui et al., 2017); on the other hand, increases microbial





activity and abundance by low N deposition (Huang et al., 2015b). But this was inconsistent with a
young *Cunninghamialanceolata* forest (Wang et al., 2017), and beneath shrubs of *H.*
*ammodendron*, soil high N content, has the opposite effect in our study site (Chen et al., 2013;
Huang et al., 2015b). What's more, the results of nonlinear regression analysis showed that higher
$R_s$ rates occurred at moderate soil $NH_4^+$-N contents (between 6.3 and 12.6 mg N / kg), while lower
$R_s$ occurred in much lower (<6.3 mg N / kg) or much higher ( >12.6 mg N / kg) soil $NH_4^+$-N
contents (Fig. 4d), but this effect of N deposition on $R_s$ is not consistent with other ecosystems
(Burton et al., 2004; Chen et al., 2013; Liu et al., 2015; Chen et al., 2017b). This is because the
desert soil is extremely limited than other ecosystem (Adams, 2003), so low N deposition
enhanced plants growth and microbial activity, but high N inhibited microbial activity and
community composition, reduced $R_s$ (Zhou et al., 2014; Huang et al., 2015b). Overall, soil
$NH_4^+$-N content was an important controlling factor for $R_s$ because microbial activity, abundance
and species diversity were regulated by soil $NH_4^+$-N content in this desert, and $R_s$ was very
sensitive to variation of N deposition.

Warming decreased $R_s$ (Fig. 2e), although not significantly (P=0.084; Table 2), which was

consistent with results from a semi-arid alfalfa-pasture of the Loess Plateau (Fang et al., 2017). In
addition, a significant decrease in $R_s$ was observed on a hot sunny day when soil moisture was
reduced, and sharply reduced $R_s$ when soil temperature reached 37 $^0C$ (Fig. 3a and 3b). This is
because: (i) microbial activity is significantly inhibited by extreme temperatures and low soil
moisture, may reduce population size by 50-80% (Sheik et al., 2011); (ii) fine root growth is
inhibited in high temperature and low soil moisture. Others have noted this phenomenon as
occurring at about 16:00 each day (Ma et al., 2013), but in our study the effect was advanced to



14:00 by warming, which may reduce carbon emission from soil to atmosphere. However, this is
not consistent with results from a tundra ecosystem, subtropical forest or alpine regions where $R_s$
was significantly increased by warming due to the limitation of soil temperature in these
ecosystems, and no significant change in soil moisture (Noh et al., 2016; Wu et al., 2016; Zhou et
al., 2016b). In addition, a significant increase in $R_s$ was found following enhanced precipitation
with warming (Fig. 3c and 3d), which indicates that soil moisture was the most important
controlling factor for $R_s$ under a warming climate. This is consistent with other studies (Chen et al.,
2017a; Zhao et al., 2017). However, statistical analysis showed that no overall significant impact
on $R_s$ was found during the experimental period by warming, although it was reduced by 9.99%.
This is because our gas samples were taken at 10:00 – 12:00 each day, when average soil
temperatures were increased about one degree. Thus mean annual $R_s$ was not sensitive to
temperature changes this small in contrast to the very significant effects of short-term diurnal
changes in soil temperature, observed between 12:00 and 17:00 (Fig. 3a and 3c). Those results
indicated that $R_s$ dependent mainly on variations of soil moisture and temperature in the context of
warming, and climate change is likely to have a very significant effect on temperate deserts.

*4.2. The interactive effects of precipitation, N deposition and warming on $R_s$*

Interactive responses of $R_s$ were much lower than those of any single-factor, but still

increased $R_s$ overall, except interaction between precipitation and high N deposition (Table 1).
This is consistent with results in dry ecosystems (Morillas et al., 2015; Martins et al., 2016), but
not with the results of a meta-analysis that precipitation and N deposition interactive experments
were a greater extent positive effect on $R_s$ (Zhou et al., 2016a). This can be explained in our study



that soil MBC or MBN were much less in interactive treatments than that of single-factor (Fig. 1d),
which showed that a number of microorganisms were much less in interactive treatments than that
of single-factor due to much stronger N effect. As we found that $R_s$ was reduced with increasing N
deposition and precipitation by as much as 4.25% in W1N2 plots (Table 1), which showed that the
inhibiting effect of soil $NH_4^+$-N content was much stronger when there was sufficient soil
moisture (Fig. 2d). This is consistent with the results in a *Populuseuphratica* community in a
desert ecosystem (He et al., 2015). This was because (i) microbial activity was inhibited by high
or low soil moisture and high soil $NH_4^+$-N or $NO_3^-$-N content (Burton et al., 2004); and (ii) high N
tent to reduce extracellular enzyme activity and the fungal population (Maris et al., 2015). In
addition, the interactive effect of the three factors on $R_s$ in this desert was much lower than
interaction of two factors of precipitation and N deposition (Table 1), and is consistent with the
results of modeled interactive effects, which showed that three-factor interactions were rare while
two-factor interactions were more common (Luo et al., 2008). Fortunately, the interactive effect of
three factors or two factors (precipitation, N deposition and warming) in this desert could largely
reduce between-year variation on $R_s$ (Table 2), which may because (i) the limits of soil moisture,
soil temperature and soil N content were relieved for key biological processes by increasing
precipitation, N deposition and warming (Huang et al., 2015a; Liu et al., 2016b); (ii) various
factors antagonistic to each other (Zhou et al., 2016a). However, the variation in the growing
season on $R_s$ can be increased by warming, elevated precipitation and N deposition because of
their dominant effects on plant growth and microbial activity (Huang et al., 2015b), but it was the
exact opposite in the non-growing season due to reduce the limit of temperature (Zeng et al.,
2016). Those results showed that $R_s$ would be reduced under interactive effect of increasing



rainfall, temperature and N deposition in the future, and took place a positive carbon-climate
feedbacks.

*4.3. Temporal variation in treatments on $R_s$ and controlling factors*
Significantly accumulated effects on $R_s$ were found by elevated N deposition rather than
alone increasing precipitation and warming (Table 2). A previous study in here has showed that $R_s$
was decreased to N addition with experimental duration (Zhou et al., 2014), which was
inconsistent with our results (Fig. 2c) because in our study relatively lower rate of N addition than
that Zhou et al. (2014), and the composition of microbial community and soil propertie were
altered gradually by long-term and high N deposition (Fig. 1c and d; Huang et al., 2015b; Zong et
al., 2017). In addition, significantly accumulated effects in the interaction between N deposition
and precipitation or warming on $R_s$ were also found (Table 2), and $R_s$ was decreased by 4.25% by
interaction between increasing precipitation and high N deposition (Table 1), which indicated that
the response of $R_s$ to N deposition largely dependent on soil moisture in desert soil. This may be
attributed to the antagonistic interaction between elevated N deposition and precipitation on $R_h$
(Zhou et al., 2016a). Those results indicated that N deposition produced strong accumulated
effects on $R_s$ in this desert, and was enhanced largely with increasing soil moisture, which would
reduce carbon emission from soil to atmosphere.
Regression analysis shows that $R_s$ exhibited a unimodal change pattern with variations of soil
$NH_4^+$-N (Fig. 4d), mositure (Fig. 4g), and temperature (Fig. 4h), and $R_s$ was signicantly postively
correlated to soil dissloved organic carbon (DOC) and pH (Fig. 4e and 4f). However, structural
equation modeling indicated that soil temperature was the most important controling factor than



soil $NH_4^+$-N and soil moisture (Fig. 5), unsupported our hypothesis, but it is consistent with most
research results (Wu et al., 2016; Zhou et al., 2016b; Chen et al., 2017a). In addition, large
inter-annual variation was observed (CV = 41.4%) during our experiment (Table 1), while the
variation of annual precipitation and air temperature were only 4.41% and 7.78%, respectively
(Table 1), but close to the CV of spring root biomass of ephemeral plants (47.14%) with 24 times
of aboveground biomass of spring ephemeral plants in 2016 than that in 2015 (Cui et al., 2017),
which indicated that the increase of $R_s$ in 2016 was mainly from the root respiration of ephemeral
plants. This is consistent with previous study that ephemeral plants mediated inter-annual variation
of carbon fluxes in this desert (Huang et al., 2015c; Liu et al., 2016). It is different from other
ecosystems where inter-annual variations of $R_s$ were mainly dependent on variations in annual
precipitation and air temperature (Gerard et al., 1999; Asensio et al., 2007; Chen et al., 2012).
Overall, our results indicate that annual variation in $R_s$ in this temperate desert is mainly
controlled by soil temperature, but between-year variation in $R_s$ is mainly controlled by ephemeral
plants.

**5. Conclusion**

Climate change and elevated N deposition play important roles in controlling $R_s$ in

temperate deserts. We found that increasing precipitation and N deposition significantly increased
$R_s$ in the Gurbantunggut Desert, but warming reduced $R_s$, mostly because of the variation of soil
moisture. In addition, we found that the interactive responses of $R_s$ was much lower to
precipitation, N deposition and warming than that any single factors. What's more, $R_s$ are mainly
mediated by soil moisture, soil temperature and soil $NH_4^+$-N content, but soil temperature are the



most important with between-year variation in $R_s$ mainly controlled by ephemeral plants. Those
results showed that $R_s$ is very sensitive to increasing precipitation, N deposition and warming, and
their interactive effects could reduce soil carbon emissions and so reduce the impacts of climate
change.

**Acknowledgments**
This work was financially supported by the Chinese National Basic Research Program
(2014CB954202), the National Natural Science Foundation of China (41603084, 41425007,
31421092) and the Ten-Thousand Talent Program (X.J. Liu).

**Inferences**
Adams, M.B.: Ecological issues related to N deposition to natural ecosystems: research needs. Environ. Int. 29,
189-199. https://doi.org/10.1016/S0160-4120(02)00179-4, 2003.
Asensio, D., Penuelas, J., Llusia, J., Ogaya, R., Filella, L: Interannual and interseasonal Soil $CO_2$ efflux and VOC
exchange rates in a Mediterranean holm oak forest in response to experimental drought. Soil Biol Biochem 39,
2471-2484. https://doi.org/10.1016/j.soilbio.2007.04.019, 2007
Batjes, N. H.: Total carbon and nitrogen in the soils of the world. Eur J Soil Sci 47, 151-163. https://
doi.org/10.1111/ejss.12114-2, 1996.
Burton, A.J., Pregitzer, K.S., Crawford, J.N., Zogg, G.P., Zak, D.R.: Simulated chronic $NO_3^-$ deposition reduces
soil respiration in northern hardwood forests. Global Change Biol 10, 1080-1091. https://doi.org/
0.1111/j.1365-2486.2004.00737.x, 2004.
Chen, J., Zhou, X.H., Hruska, T., Cao, J.J., Zhang, B.C., Liu, C., Liu, M., Shelton, S., Guo, L., Wei, Y.L., Wang,
J.F., Xiao, S., Wang, P.: Asymmetric diurnal and monthly responses of ecosystem carbon fluxes to
experimental warming. Clean-Soil Air Water 45. https://doi.org/10.1002/clen.201600557, 2017a.
Chen, S.T., Huang, Y., Zou, J.W., Shi, Y.S., Lu, Y.Y., Zhang, W., Hu, Z.H., 2012. Interannual variability in soil
respiration from terrestrial ecosystems in China and its response to climate change. Sci China Earth Sci 55,

2091-2098.

Chen, W.W., Zheng, X.H., Chen, Q., Wolf', B., Butterbach-Bahl, K., Bruggemann, N., Lin, S.: Effects of increasing
precipitation and nitrogen deposition on $CH_4$ and $N_2O$ fluxes and ecosystem respiration in a degraded steppe
in Inner Mongolia, China. Geoderma 192, 335-340. https://doi.org/10.1016/j.geoderma.2012.08.018, 2013.
Chen, Z.J., Setala, H., Geng, S.C., Han, S.J., Wang, S.Q., Dai, G.H., Zhang, J.H.,. Nitrogen addition impacts on the



emissions of greenhouse gases depending on the forest type: a case study in Changbai Mountain, Northeast
China. J Soil Sediment 17, 23-34, https://doi.org/10.1007/s11368-016-1481-7, 2017b.
Cui, X., Yue, P., Gong, Y., Li, K., Tan, D., Goulding, K., Liu, X.: Impacts of water and nitrogen addition on
nitrogen recovery in Haloxylon ammodendron dominated desert ecosystems. Sci Total Environ 601,
1280-1288. https://doi.org/10.1016/j.scitotenv.2017.05.202, 2017.
Eswaran, H., Van Den Berg, E., Reich, P.: Organic carbon in soils of the world. Soil Sci Soc Am J 57, 192-194.
https://doi:10.2136/sssaj1993.03615995005700010034x, 1993.
Fang, C., Ye, J.S., Gong, Y.H., Pei, J.Y., Yuan, Z.Q., Xie, C., Zhu, Y.S., Yu, Y.Y.: Seasonal responses of soil
respiration to warming and nitrogen addition in a semi-arid alfalfa-pasture of the Loess Plateau, China. Sci
Total Environ 590, 729-738. https://doi.org/10.1016/j.scitotenv.2017.03.034, 2017.
Feng, J.G., Wang, J.S., Ding, L.B., Yao, P.P., Qiao, M.P., Yao, S.C.: Meta-analyses of the effects of major global
change drivers on soil respiration across China. Atmos Environ 150, 181-186.
https://doi.org/10.1016/j.atmosenv.2016.11.060, 2017.
Gerard, J.C., Nemry, B., Francois, L.M., Warnant, P.: The interannual change of atmospheric $CO_2$: contribution of
subtropical ecosystems? Geophys Res Lett 26, 243-246. https://doi.org/doi: 10.1029/1998GL900269, 1999.
Gougoulias, C., Clark, J. M., Shaw, L. J.: The role of soil microbes in the global carbon cycle: tracking the below
‐ground microbial processing of plant‐derived carbon for manipulating carbon dynamics in agricultural
systems. Journal of the Science of Food and Agriculture, 94(12), 2362-2371. https://doi.org/
10.1002/jsfa.6577, 2014.
Hanson, P.J., Edwards, N.T., Garten, C.T., Andrews, J.A.: Separating root and soil microbial contributions to soil
respiration: A review of methods and observations. Biogeochemistry 48, 115-146.
https://doi.org/10.1023/A:1006244819642, 2000.
Heimann, M., Reichstein, M.:Terrestrial ecosystem carbon dynamics and climate feedbacks. Nature 451, 289-292.
https://doi.org/doi:10.1038/nature06591, 2008.
He, X., Lv, G., Qin, L., Chang, S., Yang, M., Yang, J., Yang, X.: Effects of simulated nitrogen deposition on soil
respiration in a Populus euphratica community in the Ebinur Lake area, a desert ecosystem of northwestern
China. PloS one 10, 1-16. https://doi.org/10.1371/journal.pone.0137827, 2015.
Huang, G., Li, Y., Su, Y.G.: Effects of increasing precipitation on soil microbial community composition and soil
respiration in a temperate desert, Northwestern China. Soil Biol Biochem 83, 52-56.
https://doi.org/10.1016/j.soilbio.2015.01.007, 2015a..
Huang, G., Cao, Y.F., Wang, B., Li, Y.; Effects of nitrogen addition on soil microbes and their implications for soil
C emission in the Gurbantunggut Desert, center of the Eurasian Continent. Sci Total Environ 515, 215-224,
https://doi.org/10.1016/j.scitotenv.2015.01.054, 2015b.
Huang, G., Li, Y., Padilla, F.M.: Ephemeral plants mediate responses of ecosystem carbon exchange to increased
precipitation in a temperate desert. Agr Forest Meteorol 201, 141-152,
https://doi.org/10.1016/j.agrformet.2014.11.011, 2015c.
IPCC, 2013. Climate Change 2013: the Physical Science Basis. Contribution of Working Group I to the Fifth
Assessment Report of the Intergovernmental Panel on Climate Change. Cambridge University Press,
Cambridge, United Kingdom and New York, NY, USA, 1535 pp.



Jiang, X., Cao, L., Zhang, R.:. Changes of labile and recalcitrant carbon pools under nitrogen addition in a city
lawn soil. J. Soils Sediments 14, 515–524. https://doi.org/10.1007/s11368-013-0822-z, 2014.
Lafleur, P.M., Humphreys, E.R.,: Spring warming and carbon dioxide exchange over low Arctic tundra in central
Canada. Global Change Biol 14, 740-756, https://doi.org/10.1111/j.1365-2486.2007.01529.x, 2008.
Liu, C., Wang, K., Zheng, X.: Responses of $N_2O$ and $CH_4$ fluxes to fertilizer nitrogen addition rates in an irrigated
wheat-maize      cropping      system      innorthern      China.      Biogeosciences      9,      839-850,
https://doi.org/10.5194/bg-9-839-2012, 2012.
Li, G., Kim, S., Han, S.H., Chang, H., Son, Y., 2017. Effect of soil moisture on the response of soil respiration to
open-field experimental warming and precipitation manipulation. Forests 8.
Liu, L.T., Hu, C.S., Yang, P.P., Ju, Z.Q., Olesen, J.E., Tang, J.W.: Effects of experimental warming and nitrogen
addition on soil respiration and $CH_4$ fluxes from crop rotations of winter wheat-soybean/fallow. Agr Forest
Meteorol 207, 38-47. https://doi.org/10.1016/j.agrformet.2015.03.013, 2015.
Liu, T., Xu, Z.Z., Hou, Y.H., Zhou, G.S.: Effects of warming and changing precipitation rates on soil respiration
over two years in a desert steppe of northern China. Plant Soil 400, 15-27. https://doi.org/
https://doi.org/10.1007/s11104-015-2705-0, 2016a.
Liu, R., Cieraad, E., Li, Y., Ma, J.: Precipitation pattern determines the inter-annual variation of herbaceous layer
and carbon fluxes in a phreatophyte-dominated desert ecosystem. Ecosystems 19, 601-614.
https://doi.org/10.1007/s10021-015-9954-x, 2016.
Liu, L.L., Wang, X., Lajeunesse, M.J., Miao, G.F., Piao, S.L., Wan, S.Q., Wu, Y.X., Wang, Z.H., Yang, S., Li, P.,
Deng, M.F.: A cross-biome synthesis of soil respiration and its determinants under simulated precipitation
changes. Global Change Biol 22, 1394-1405. https://doi.org/ 10.1111/gcb.13156, 2016c.
Liu, Y., Li, X., Zhang, Q., Guo, Y., Gao, G., Wang, J.: Simulation of regional temperature and precipitation in the
past      50      years      and      the      next      30      years      over      China.      Quatern      Int      212,      57-63
https://doi.org/10.1016/j.quaint.2009.01.007, 2010a.
Luo, C.Y., Wang, S.P., Zhao, L., Xu, S.X., Xu, B.R.B.Y., Zhang, Z.H., Yao, B.Q., Zhao, X.Q.: Effects of land use
and nitrogen fertilizer on ecosystem respiration in alpine meadow on the Tibetan Plateau. J Soil Sediment 17,
1626-1634, https://doi.org/10.1016/j.quaint.2009.01.007, 2017.
Luo, Y., Gerten, D., Le Maire, G., Parton, W.J., Weng, E., Zhou, X., Keough, C., Beier, C., Ciais, P., Cramer, W.:
Modeled interactive effects of precipitation, temperature, and $CO_2$ on ecosystem carbon and water dynamics
in      different      climatic      zones.      Global      Change      Biol      14,      1986-1999.      https://doi.org/
10.1111/j.1365-2486.2008.01629.x, 2008.
Ma, J., Wang, Z.-Y., Stevenson, B.A., Zheng, X.-J., Li, Y.: An inorganic $CO_2$ diffusion and dissolution process
explains negative $CO_2$ fluxes in saline/alkaline soils. Sci Rep-Uk 3. https://doi.org/10.1038/srep02025, 2013.
Maris, S.C., Teira-Esmatges, M.R., Arbonés, A., Rufat, J.: Effect of irrigation, nitrogen application, and a
nitrification inhibitor on nitrous oxide, carbon dioxide and methane emissions from an olive (Olea europaea
L.) orchard. Sci. Total Environ. 538,966–978, https://doi.org/10.1016/j.scitotenv.2015.08.040, 2015.
Martins, C.S.C., Macdonald, C.A., Anderson, I.C., Singh, B.K.: Feedback responses of soil greenhouse gas
emissions to climate change are modulated by soil characteristics in dryland ecosystems. Soil Biol Biochem
100, 21-32, https://doi.org/10.1016/j.soilbio.2016.05.007,2016.





Morillas, L., Duran, J., Rodriguez, A., Roales, J., Gallardo, A., Lovett, G.M., Groffman, P.M.: Nitrogen supply
modulates the effect of changes in drying-rewetting frequency on soil C and N cycling and greenhouse
gas exchange. Global Change Biol 21, 3854-3863, https://doi.org/10.1111/gcb.12956, 2015.
Noh, N.J., Kuribayashi, M., Saitoh, T.M., Nakaji, T., Nakamura, M., Hiura, T., Muraoka, H.: Responses of soil,
heterotrophic, and autotrophic respiration to experimental open-field soil warming in a cool-temperate
deciduous forest. Ecosystems 19, 504-520, https://doi.org/10.1007/s10021-015-9948-8, 2016.
Sheik, C.S., Beasley, W.H., Elshahed, M.S., Zhou, X., Luo, Y., Krumholz, L.R.: Effect of warming and drought on
grassland microbial communities. The ISME Journal 5, 1692-1700, https://doi.org/ 10.1038/ismej.2011.32,
493     2011.

Skopp, J., Jawson, M. D., Doran, J. W.: Steady-state aerobic microbial activity as a functionof soil watercontent.
Soil Sci Soc Am J54, 1619-1625, https://doi.org/10.2136/sssaj1990.03615995005400060018x, 1990.
Strong, A.L., Johnson, T.P., Chiariello, N.R., Field, C.B.: Experimental fire increases soil carbon dioxide efflux in
a grassland long-term multifactor global change experiment. Global Change Biol 23, 1975-1987.
https://doi.org/10.1111/gcb.13525, 2017.
Wang, Q.K., Zhang, W.D., Sun, T., Chen, L.C., Pang, X.Y., Wang, Y.P., Xiao, F.M.: N and P fertilization reduced
soil autotrophic and heterotrophic respiration in a young Cunninghamia lanceolata forest. Agr Forest
Meteorol 232, 66-73, https://doi.org/10.1016/j.agrformet.2016.08.007, 2017.
Wu, C.S., Liang, N.S., Sha, L.Q., Xu, X.L., Zhang, Y.P., Lu, H.Z., Song, L., Song, Q.H., Xie, Y.N., 2016.
Heterotrophic respiration does not acclimate to continuous warming in a subtropical forest. Sci Rep-Uk 6.
Wu, Z.T., Dijkstra, P., Koch, G.W., Penuelas, J., Hungate, B.A.: Responses of terrestrial ecosystems to temperature
and precipitation change: a meta-analysis of experimental manipulation. Global Change Biol 17, 927-942,
https://doi.org/10.1111/j.1365-2486.2010.02302.x, 2011.
Yang, G., Wang, M., Chen, H., Liu, L.F., Wu, N., Zhu, D., Tian, J.Q., Peng, C.H., Zhu, Q.A., He, Y.X.: Responses
of $CO_2$ emission and pore water DOC concentration to soil warming and water table drawdown in Zoige
Peatlands. Atmos Environ 152, 323-329. https://doi.org/10.1016/j.atmosenv.2016.12.051, 2017.
Zeng, X.H., Song, Y.G., Zeng, C.M., Zhang, W.J., He, S.B.: Partitioning soil respiration in two typical forests in
semi-arid regions, North China. Catena 147, 536-544. https://doi.org/10.1016/j.catena.2016.08.009, 2016.
Zhang, N., Liu, W., Yang, H., Yu, X., Gutknecht, J.L.M., Zhang, Z., Wan, S., Ma, K.: Soil microbial responses to
warming and increased precipitation and their implications for ecosystem C cycling. Oecologia 173,
1125-1142. https://doi.org/ 10.1007/s00442-013-2685-9, 2013.
Zhang, X.L., Tan, Y.L., Zhang, B.W., Li, A., Daryanto, S., Wang, L.X., Huang, J.H.: The impacts of precipitation
increase and nitrogen addition on soil respiration in a semiarid temperate steppe. Ecosphere 8. https:
//doi.org/ 10.1002/ecs2.1655, 2017.
Zhang, W. Liu,C.Y., Zheng, X.H., Fu, Y.F., Hu, X,X., Cao,G.M., Klaus Butterbach-Bahl.: The increasing
distribution area of zokor mounds weaken greenhouse gas uptakes by alpine meadows in the Qinghai-Tibetan
Plateau. Soil Biol Biochem, 71: 105-112. https://doi.org/10.1016/j.soilbio.2014.01.005, 2014.
Zhao, Z.Z., Dong, S.K., Jiang, X.M., Liu, S.L., Ji, H.Z., Li, Y., Han, Y.H., Sha, W.: Effects of warming and
nitrogen deposition on $CH_4$, $CO_2$ and $N_2O$ emissions in alpine grassland ecosystems of the Qinghai-Tibetan
Plateau. Sci Total Environ 592, 565-572. https://doi.org/10.1016/j.scitotenv.2017.03.082, 2017.





Zhong, Y.Q.W., Yan, W.M., Shangguan, Z.P.: The effects of nitrogen enrichment on soil $CO_2$ fluxes depending on
temperature and soil properties. Global Ecol Biogeogr 25, 475-488. https://doi.org/10.1111/geb.12430, 2016.
Zhou, L.Y., Zhou, X.H., Shao, J.J., Nie, Y.Y., He, Y.H., Jiang, L.L., Wu, Z.T., Bai, S.H.: Interactive effects of
global change factors on soil respiration and its components: a meta-analysis. Global Change Biol 22,
3157-3169. https://doi.org/ 10.1111/gcb.13253, 2016a.
Zhou, X.B. and Zhang, Y.M.: Seasonal pattern of soil respiration and gradual changing effects of nitrogen addition
in a soil of the Gurbantunggut Desert, northwestern China. Atmos Environ, 85: 187-194.
https://doi.org/10.1016/j.atmosenv.2013.12.024, 2014.
Zhou, Y.M., Hagedorn, F., Zhou, C.L., Jiang, X.J., Wang, X.X., Li, M.H.: Experimental warming of a mountain
tundra increases soil $CO_2$ effluxes and enhances $CH_4$ and $N_2O$ uptake at Changbai Mountain, China. Sci
Rep-Uk 6. https://doi.org/10.1038/srep21108, 2016b.
Zhu, J., Kang, F.F., Chen, J., Cheng, X.Q., Han, H.R.: Effect of nitrogen addition on soil respiration in a Larch
Plantation. Pol J Environ Stud 26, 1403-1412. https://doi.org/10.15244/pjoes/67687, 2017.
Zong, N., Shi, P.L., Chai, X., Jiang, J., Zhang, X.Z., Song, M.H.: Responses of ecosystem respiration to nitrogen
enrichment and clipping mediated by soil acidification in an alpine meadow. Pedobiologia 60, 1-10.
https://doi.org/10.1016/j.pedobi.2016.11.001, 2017.






















**Fig. 1** Comparative effects of warming by Open Topped Chambers, increased precipitation, and N
deposition on soil temperature (a), soil moisture (b) at 5 and 20 cm depth; soil organic carbon
(SOC), $NH_4^+$-N content, $NO_3^-$-N content and pH (c), dissolved organic carbon (DOC), microbial
biomass carbon (MBC) and microbial biomass nitrogen (MBN, d). The data are mean $\pm$ SE, n = 4
in c and d, different letter indicate significant effect at $P < 0.05$.

**Fig. 2** Variation in rainfall (mm, a) and air temperature (℃, a) from September 2014 to October
2016 at the Gurbantunggut Desert, and the response of $R_s$ (mean, n = 4) to precipitation (b), N
deposition (c-d) and warming (e). W0 and W1 indicate under ambient precipitation (without water
addition) and 60 mm $yr^{-1}$ precipitation addition; N0, N1, and N2 indicate 0, 30 and 60 kg N $ha^{-1}$
$yr^{-1}$ nitrogen addition; while W1N0, W1N1, and W1N2 indicate 0, 30 and 60 kg N $ha^{-1}$ $yr^{-1}$
nitrogen addition under 60 mm $yr^{-1}$ precipitation addition; W1N1T1, W0N0T1 and W0N0T0
indicate the interaction between increasing precipitation (60 mm $yr^{-1}$), N deposition (30 kg N $ha^{-1}$
$yr^{-1}$) and warming by OTCs, warming alone (without increasing precipitation and N deposition)
and control plots, respectively. Black arrows indicate simulated precipitation (5 mm per time) and
N deposition (0.25 or 0.5 g N $m^{-2}$ per time). Each point represents the mean of four replications
(chambers). Standard deviations for $R_s$ are not showed for figure clarity.

**Fig. 3** Post-rainfall diurnal variation in $R_s$ (mean $\pm$ SE, n = 4, b) with variation in soil temperature
and soil moisture (a), and a sunny day variation in $R_s$ (mean $\pm$ SE, n = 4, d) with variation in soil
temperature (T5, T20, c) and soil moisture at 5 (W5) or 20 (W20) cm depth caused by warming in
open topped chambers (OTCs). Positive values indicate increment by warming, and negative
values indicate decline. A red straight line indicates the average value of $R_s$ inside the OTCs in (b)
and (d), and a green straight line represents the average value of $R_s$ out of OTCs in (b) and (d).
Red *, ** and *** indicate significant effect at $P < 0.05$, $P < 0.01$, and $P < 0.001$, respectively.

**Fig. 4** The relationship of soil respiration with microbial biomass carbon (MBC, a); microbial
biomass carbon (MBN, b); soil $NO_3$-N (c); $NH_4^+$-N content (d); soil dissolved organic carbon
(DOC, e); pH (f); soil moisture (g) and soil temperature (h).



**Fig. 5** Structure equation modeling (SEM) test the multivariate (soil moisture, soil temperature,
soil $NH_4^+$-N content, DOC and pH) effects on $R_s$ (n=34). Single-headed arrows show that the
effect of different key controls on $R_s$ were analyzed. The green arrows indicated positive effects,
and red arrows showed negative effects. And the width of the arrows indicate the strength of the
relationship. The numbers are standardized path coefficients, which can show the importance of
the variables in the model. Goodness-of-fit statistics for the model are shown below the model. *
indicate significant effect at $P < 0.05$.























**Fig. 1**

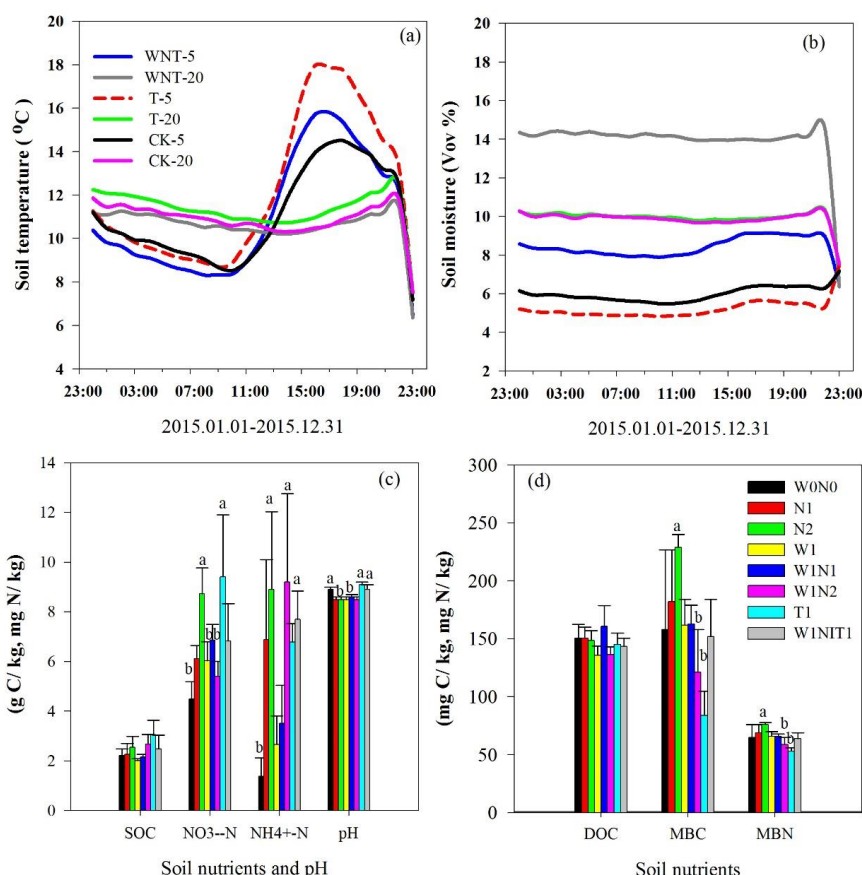















**Fig. 2**





**Fig. 3**




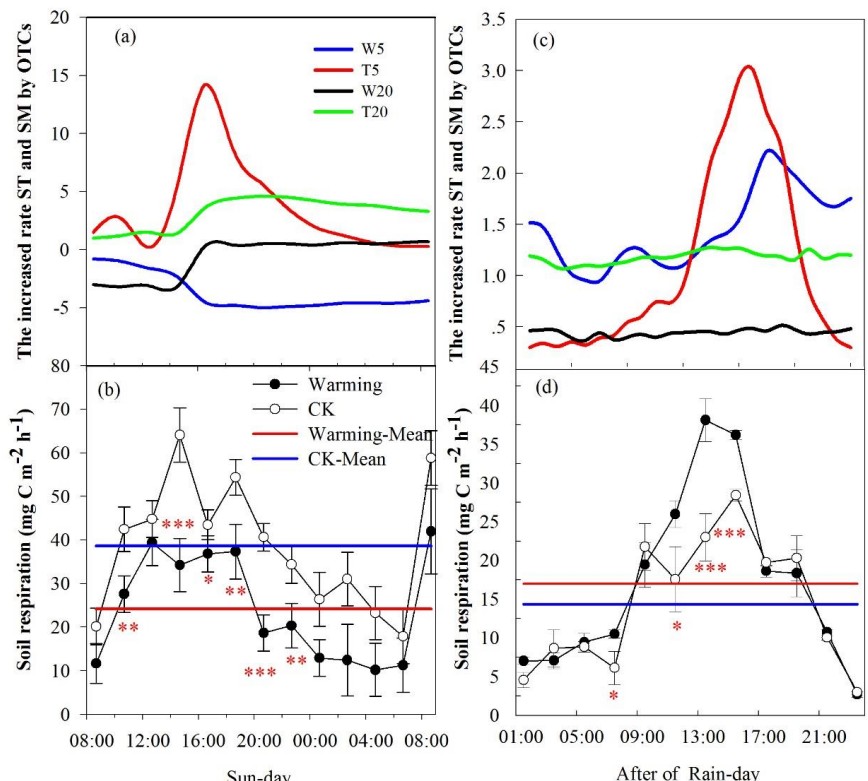






**Fig. 4**

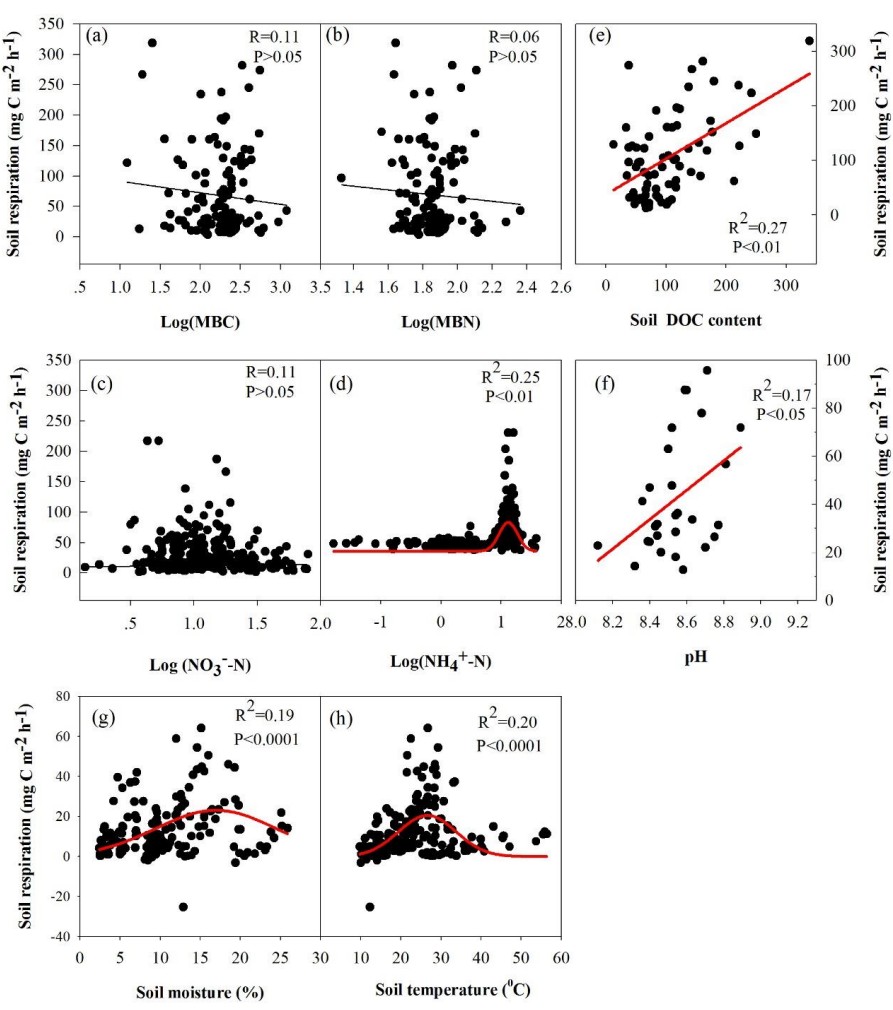












**Fig. 5**

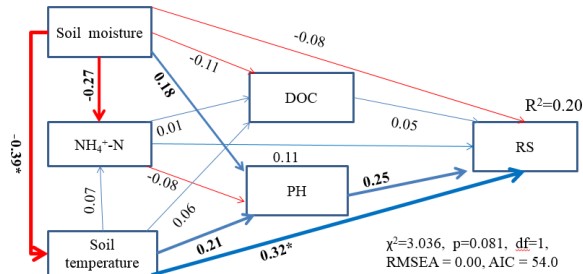




























**Table 1** The annual, growing season (GS), non-growing season (NGS), and between-year fluxes
and variation of soil respiration ($R_s$) in September 2014 to September 2016 (mean ±SE), including
the contribution of GS and NGS, and the treatment effect. The positive values stand for increase
$R_s$, and the negative value stand for reduced $R_s$.

| Treatments | Prec. mm | temp. $^0$C | $R_s$ rate (kg C ha$^{-1}$) | | | | | | | |
|---|---|---|---|---|---|---|---|---|---|---|
| | | | W0N0 | W0N1 | W0N2 | W1N0 | W1N1 | W1N2 | W1N1T1 | T1 |
| Annual | 175.75 | 4.63 | 1090.11 | 1338.26 | 1299.41 | 1450.78 | 1304.77 | 1043.77 | 1196.84 | 981.19 |
| | ±7.75 | ±0.36 | ±450.78 | ±599.12 | ±537 | ±543.70 | ±383.29 | ±233.23 | ±334.31 | ±371.34 |
| CV (%) | 4.41 | 7.78 | 41.35 | 44.77 | 41.33 | 37.48 | 29.37 | 22.34 | 27.93 | 37.85 |
| Treatments effect (%) | --- | --- | --- | 22.76 | 19.20 | 33.09 | 19.69 | -4.25 | 9.79 | -9.99 |
| Growing season | | | | | | | | | | |
| 2014.9-2015.8 | 120.5 | 14.67 | 508.30 | 561.95 | 570.38 | 650.66 | 669.93 | 562.04 | 561.10 | 425.35 |
| 2015.9-2016.8 | 114.5 | 21.18 | 1220.27 | 1546.62 | 1506.88 | 1641.62 | 1274.93 | 1052.22 | 1183.86 | 1089.51 |
| CV (%) | 3.61 | 25.68 | 46.18 | 52.03 | 50.96 | 48.30 | 32.79 | 33.21 | 50.47 | 62.00 |
| Non-Growing season | | | | | | | | | | |
| 2014.9-2015.8 | 47.5 | -6.13 | 131.03 | 177.09 | 192.03 | 256.43 | 251.55 | 248.50 | 301.43 | 184.49 |
| 2015.9-2016.8 | 69 | -11.2 | 320.62 | 390.84 | 329.54 | 352.85 | 413.12 | 224.79 | 347.29 | 262.02 |
| CV (%) | 26.10 | 41.37 | 59.37 | 53.23 | 37.29 | 22.38 | 34.38 | 7.09 | 10.00 | 24.82 |
| NGS Contribution | --- | --- | 20.65 | 22.07 | 21.57 | 22.98 | 25.89 | 24.13 | 27.10 | 22.80 |

















**Table 2** Tests of significance of year (Y), warming (T), precipitation (W) and nitrogen addition
(N) on soil respiration ($R_s$) by multivariate ANOVA (F and P values). The accumulated effect of
precipitation, N deposition and warming on $R_s$ in 2015 and 2016 (F and P values) as assessed by
repeated measures ANOVA. *, ** and *** indicate significant effect at P < 0.05, P < 0.01, and P <
0.001, respectively.

| Three-way ANOVA | n | F | P |
|---|---|---|---|
| Y | 2 | 26.171 | <0.001*** |
| N | 424 | 7.709 | <0.001*** |
| W | 565 | 17.124 | <0.001*** |
| W×N | 424 | 9.392 | <0.001*** |
| W×Y | 424 | 6.899 | <0.001*** |
| N×Y | 424 | 5.561 | 0.004** |
| Y×W×N | 424 | 5.963 | 0.003** |
| T | 424 | 2.320 | 0.084 |
| T×Y | 424 | 0.536 | 0.464 |
| Repeated measures ANOVA | | | |
| Y | 2 | 30.487 | <0.000*** |
| N | 383 | 12.887 | <0. 000*** |
| W | 281 | 2.934 | 0.087 |
| T | 142 | 0.965 | 0.326 |
| W×N | 281 | 12.755 | <0.000*** |
| T×W×N | 281 | 39.927 | <0.000*** |
