# Peer review of "Impact of elevated precipitation, nitrogen deposition and warming on soil"

_Biogeosciences, 2017_

## Referee Comment (RC1) · Anonymous Referee #1 · 5 Dec 2017

The effects of elevated precipitation, N deposition and warming on soil respiration was analyzed in a temperature desert based on 2-years data. Its valuable to promote the research on the response of soil respiration to climate change in dry land. But this manuscript needs major revision before publication.

detailed comments: 1. Fig 3 showed the diurnal variation of Rs during one sun day and one post-rain day, so the diurnal pattern of Rs in Fig 3 may can not represent the diurnal pattern of Rs across the whole year. If not, the measured Rs during 10:00-12:00 may show large difference with the daily average of Rs and further failed to show the effect of treatment on Rs for everyday in 2014-2016. It may be better to show the

diurnal pattern of Rs at different seasons. 2. all gas samples were taken at 10:00-12:00 in everyday, however, the warming effect on soil temperature is not obvious during this sampling time (the obvious warming effect on soil temperature occurred at midday and afternoon time, fig 3a). So the samples during 10:00-12:00 in this study may failed to catch the real warming effects on Rs.

---

## Referee Comment (RC2) · Anonymous Referee #2 · 18 Jan 2018

This manuscript studied the effects of elevated precipitation, N deposition and warming on soil respiration in a temperate desert. This study was well designed, the manuscript was also well written and the results are interesting, which have important implications on the climate change feedback of soil respiration in the temperate desert. I recommend this manuscript to be accepted with minor revision. The major comments are as follows.

1. Line 134, what is the principle for the increased temperature caused by OTC? How much temperature can be increased by this OTC?

2. Lines 141-144, did the Rs measured in this study also include the above-ground

respiration of the plants? It seems that there were no measures to exclude the above-ground respiration.

3. Lines 166-167, how to calculate the interactive effects of precipitation, N deposition and warming on Rs?

4. Line 195, it seems soil moisture was mainly affected by the elevated precipitation other than the interaction of precipitation, N deposition and warming.

5. Fig.1 a and b, what were the seasonal variations for soil T and moisture?

6. Fig. 4f, why the data number in Fig.4f is less than other figures in Fig. 4?

Lines 232-239, did the thresholds be calculated using statistical method?

Some minor comments: 7. Line 138, please use "the same as".

8. Page 6, please give the exact year when the experiments were conducted.

9. Lines 158-159, references for the MBC and MBN measurement should be given.

10. Line 160, can soil pH be measured using potassium dichromate method? It must be a mistake.

11. Line 161, can't find the reference of Yue et al. (2016) in the reference list.

12. Fig.1 c and d, these figures should be enlarged. It's hard to see.

---

## Author Comment (AC1) · 6 Feb 2018

Reviewer 1 The effects of elevated precipitation, N deposition and warming on soil respiration was analyzed in a temperature desert based on 2-years data. Its valuable to promote the research on the response of soil respiration to climate change in dry land. But this manuscript needs major revision before publication. Response: Thanks very much for your revision. We accept and have made the changes requested.

Detailed comments: 1. Fig 3 showed the diurnal variation of Rs during one sun day and one post-rain day, so the diurnal pattern of Rs in Fig 3 may can not represent the diurnal pattern of Rs across the whole year. If not, the measured Rs during 10:00-

12:00 may show large difference with the daily average of Rs and further failed to show the effect of treatment on Rs for everyday in 2014-2016. It may be better to show the diurnal pattern of Rs at different seasons. Response: Thank you for your comment. Diurnal variations of Rs were only measured from March to September in 2015, March, April and July in 2016 (Fig.1S.). Firstly, we have also corrected a 'wrong' description in Fig 3 in the original manuscript. Now the Fig 3a and b have shown the diurnal variation of Rs during extreme drought (continuous high temperature drought) rather than one normal sunny day, and the Fig 3c and d have shown the diurnal variation of Rs during an extreme wet day (with daily precipitation 33 mm) rather than one small post-rainy day. We found that the diurnal average of Rs were closed to the observed value during 10:00-12:00 from daily change observations in 2015 and 2016, except in July 2015, (Fig.1S. J). Therefore, this supported the effect of treatments. Please see lines 147-149. Thanks again.

2. All gas samples were taken at 10:00-12:00 in everyday, however, the warming effect on soil temperature is not obvious during this sampling time (the obvious warming effect on soil temperature occurred at midday and afternoon time, fig 3a). So the samples during 10:00-12:00 in this study may failed to catch the real warming effects on Rs. Response: Yes, a varying effect on Rs was observed by warming in Fig 3. The data came from extreme precipitation and drought events mainly, which may overestimate the warming effect on Rs. The Rs can be inhibited at high temperature and low humidity, a common phenomenon in the summer; and warming can reinforce this effect in Fig 1S. j, l, n and t. However, our results could represent the warming effects on Rs in spring (e.g. April) and autumn although high temperature reduced Rs, because the observed values on the diurnal average of Rs in warming plots are close to the real values of Rs during 10:00-12:00, except some extreme precipitation and drought events that in summer. So the samples during 10:00-12:00 in this study could catch the mean warming effects on Rs as a whole. We also have made further discussion in the revised text. Please see lines 299-307.

Reviewer2 Anonymous Referee #2 This manuscript studied the effects of elevated precipitation, N deposition and warming on soil respiration in a temperate desert. This study was well designed, the manuscript was also well written and the results are interesting, which have important implications on the climate change feedback of soil respiration in the temperate desert. I recommend this manuscript to be accepted with minor revision. The major comments are as follows. 1. Line 134, what is the principle for the increased temperature caused by OTC? How much temperature can be increased by this OTC? Response: The principle of an OTC warming is to heat the air in an OTC system through solar radiation, and the OTC system has an effect of windshield, so the temperature in OTC was increased. The air temperature was increased by about 1 0C on average, and the average annual soil temperatures at 5 and 20 cm depth were significantly increased by 4.41 and 3.67 0C, respectively (Fig. 1a). An additional sentence was added in the revision. Please see lines 195-196. 2. Lines 141-144, did the Rs measured in this study also include the above-ground respiration of the plants? It seems that there were no measures to exclude the aboveground respiration. Response: Yes, Rs measured in this study also include parts of the above-ground respiration of the plants but only from April to May. Because ephemeral plants grow only during this period. In addition, the ephemeral plants are very sparse, and cover only 20-30% of total area, so the above-ground respiration of the plants was relatively weak.

3. Lines 166-167, how to calculate the interactive effects of precipitation, N deposition and warming on Rs? Response: The interactive effects of precipitation, N deposition and warming on Rs were calculated by the treatments between W1N1T1 plots and W0N0 plots. However, there were lack of interactive effects of N deposition and warming, so the interactive effects of precipitation, N deposition and warming on Rs were not calculated by repeated measures of variance analysis.

4. Line 195, it seems soil moisture was mainly affected by the elevated precipitation other than the interaction of precipitation, N deposition and warming. Response:

Agreed and corrected, please see sentence in lines 198-199.

5. Fig.1 a and b, what were the seasonal variations for soil T and moisture? Response: The Fig.1a and b showed that the diurnal variation for soil T and moisture. We have added the seasonal variations for soil T and moisture in Fig 2b.

6. Fig. 4f, why the data number in Fig.4f is less than other figures in Fig. 4? Response: This is because soil pH in soil samples were only measured in several times.

7. Lines 232-239, did the thresholds be calculated using statistical method? Some minor comments: Response: The thresholds were re-analyzed or calculated using Nonlinear Regression (3D, Gaussian and Plane) as in Fig 2S and Fig 4f. We found that Rs was inhibited at high temperature and low humidity (soil temperature > 26.5 0C and soil moisture < 4.2 %), and low temperature and high humidity (soil temperature <2.7 0C and soil moisture >15.9 %). However, moderate soil temperature and moisture increased Rs (Fig. 2S). Therefore, it can be summarized as the response characteristics of Rs under different temperature and humidity ranges rather than the 'true' threshold. We have corrected a 'wrong' description on thresholds in the text, because of no particular accurate threshold by current statistical analysis. Please see lines 234-238.

8. Line 138, please use "the same as". Response: Agreed and corrected. Please see lines 139-140.

9. Page 6, please give the exact year when the experiments were conducted. Response: Agreed and done, please see line 114.

10. Lines 158-159, references for the MBC and MBN measurement should be given. Response: A reference has been added (line 400). Please see lines 160-161.

11. Line 160, can soil pH be measured using potassium dichromate method? It must be a mistake. Response: Thank you for correcting this mistake. We have corrected the wrong description. Please see lines 163-164.

12. Line 161, can't find the reference of Yue et al. (2016) in the reference list. Response: The reference of Yue et al. (2016) has been added in the reference list. Please see lines 517-518.

13. Fig.1 c and d, these figures should be enlarged. It's hard to see. Response: Agreed and done as suggested. s 299-307.

Please also note the supplement to this comment:
https://www.biogeosciences-discuss.net/bg-2017-465/bg-2017-465-AC1-supplement.zip

[Figure]

Increase in ST and SM by OTCs
Increase at SM5
Increase at ST5
(a)
(b)
(c)
(d)
(e)
(f)
(g)
(h)
(i)
(j)
$R_S$ rate (mg C m$^{-2}$h$^{-1}$)
March 2015
April 2015
May 2015
June 2015
July 2015
(k)
(l)
(m)
(n)
(o)
(p)
(q)
(r)
(s)
(t)
OTCs
CK
Mean value in CK
Mean value in OTCs
August 2015
September 2015
March 2016
April 2016
July 2016

Fig. 1.

[Figure]

HHLT $R^2$=0.38 P<0.001
MTM $R^2$=0.26 P<0.0001
HTLH $R^2$=0.45 P<0.001
$R_s$ (mg C $m^{-2}$ $h^{-1}$)

Soil temperature ($^0$C)

Soil moisture (%)

Fig. 2.